# Gender Disparity in Host Responses to Hepatitis B-Related Hepatocellular Carcinoma: A Case Series

**DOI:** 10.3390/vaccines9080838

**Published:** 2021-07-30

**Authors:** Rukaiya Bashir Hamidu, Divya M. Chalikonda, Hie-Won Hann

**Affiliations:** 1Department of Medicine, Thomas Jefferson University Hospital, Philadelphia, PA 19107, USA; Rukaiya.Bashir-Hamidu@jefferson.edu; 2Department of Medicine, Division of Gastroenterology and Hepatology, Thomas Jefferson University Hospital, Philadelphia, PA 19107, USA; divya.chalikonda@jefferson.edu; 3Liver Disease Prevention Center, Division of Gastroenterology and Hepatology, Thomas Jefferson University Hospital, Philadelphia, PA 19107, USA

**Keywords:** hepatitis B virus, chronic hepatitis B, hepatocellular carcinoma, stress, gender disparity, sex difference, survival, prognosis, estrogens, androgens

## Abstract

Chronic hepatitis B virus (HBV) infection is one of the most common causes of hepatocellular carcinoma (HCC), a malignant tumor with high mortality worldwide. One remarkable clinical feature of HBV-related HCC is that the risk of development is higher in males and postmenopausal females compared to other females. Increasing evidence also indicates that the prognosis of HBV-associated HCC may involve gender disparity, with females having more favorable outcomes. The proposed mechanism of this gender disparity is thought to be complex and multifactorial. Attributions have been made to gender differences in behavioral risk factors, host stress, immune response, psychology, metabolic risk factors, tumor biology, and hormonal factors. Gender disparities in hormonal factors and stress with consequent incited inflammation and hepatocarcinogenesis in HBV-related HCC is a particularly burgeoning area of investigation. Clarifying these mechanisms could provide insight into HBV-related HCC pathogenesis, and potentially provide a target for prevention and treatment of this disease. Reported herein is a case series involving two families affected by vertically transmitted chronic hepatitis B, longitudinally observed over multiple decades, with family members demonstrating discordant outcomes related to HCC, with worse outcomes among affected males. As a supplement to this case, we review the currently available literature on gender differences in outcomes from HBV-related HCC. In reporting this case series, we aim to add our important observation to the current literature and highlight the need for further research in the mechanisms involved in gender disparity in the prognosis of HBV-related HCC.

## 1. Introduction

The hepatitis B virus (HBV) is a non-cytopathic pathogen that can establish chronic infection in the liver [1,2]. Since the advent of the effective HBV vaccine and the widespread implementation of HBV vaccination programs, the incidence of HBV infection has decreased worldwide [3,4]. Nonetheless, the burden of HBV infection persists; it roughly affects 2 billion people worldwide, among which approximately 300 million people are chronically infected [1,2]. Chronic hepatitis B (CHB) is a common cause of both liver cirrhosis and hepatocellular carcinoma (HCC) [1,2,5]. Given its known carcinogenic properties, HBV accounts for 25% of HCC cases in developed countries and nearly 60% of cases in developing countries [1,2,6]. Clinical factors that determine the outcomes of chronic HBV infection, and HCC, thus carry substantial public health importance [2]. Illumination of this constellation of risk factors and understanding the HBV-related hepatocarcinogenesis remains an essential goal of HBV research [1].

There has been significant progress in the understanding of viral chemistry-related hepatocarcinogenesis and as a result, advances have been made in targeted anti-HBV therapy, which has ultimately decreased the morbidity and mortality associated with chronic HBV infection [5]. The incidence of HBV-related HCC has also decreased since the advent of HBV vaccine and antiviral therapy [4,7,8,9,10]. Nonetheless, despite successful treatment of HBV infection with antiviral therapy, persistent risk for HCC has been reported [11,12,13,14,15,16]. This is a phenomenon that has also been well documented at our high-volume hepatitis B center. This suggests that factors beyond the genetic make-up of the virus, and therefore response to target therapy with antivirals, influence the outcome of CHB infection. On the contrary, host factors may play a larger role in the development and prognosis of HBV-related HCC. This idea has spiked wide interest in the scientific community [2,17].

It is well established that compared to women, men have a higher incidence of HCC across varied etiologies, including HBV [6,17,18,19,20]. This supports the potential role for gender-associated differences in risk factor exposures as well as sex-based biological differences in disease progression [20]. The role of this gender disparity as a prognostic indicator in HCC has gained increased recognition [6,17,18,21].

Presented herein is a case series involving two families affected by vertically transmitted CHB, longitudinally observed over multiple decades, with family members demonstrating discordant outcomes related to HCC. Worse outcomes were observed among affected males.

## 2. Case Series

### 2.1. Family 1

In 1987, during a community HBV screening, an Asian mother and her three adult children (two older brothers and a younger sister) were found to be positive for hepatitis B surface antigen (HBsAg) with HBV genotype C, suggesting perinatal infection. None of them had human immunodeficiency virus (HIV) or hepatitis C virus (HCV) co-infections. Upon longitudinal observation, all four patients went on to develop chronic hepatitis B. Interestingly, their course of CHB-related illness over the following decades was observed to be different.

After 14 years without complications or antiviral therapy, in the year 2000 and at 64 years old (y.o), the mother developed fatigue and presented to our institution for a new patient visit. Further evaluation revealed an alpha-fetoprotein level of 6.7 (normal < 6.1 ng/mL), alanine aminotransferase (ALT) of 29 (normal < 45 IU/L), and an abdominal magnetic resonance imaging (MRI) with a 2 × 1.5 cm lesion consistent with HCC in the absence of evidence of cirrhosis or portal hypertension. She was started on antiviral therapy with Lamivudine, underwent microwave tumor ablation, and remained well with undetectable HBV DNA levels. However, she went on to have two HCC recurrences (at the age of 69 and at the age of 82), both managed with loco-regional tumor ablation and continuation of antiviral therapy. Lamivudine was later switched to Tenofovir disoproxil fumarate (TDF). Thereafter, her three children established care at our institution. The mother and her children denied significant alcohol consumption. The children were found to be in low viremic stages with minimal signs of hepatic inflammation, and antiviral therapy with TDF was started in 2008 in light of their mother’s multiple HCCs. Antiviral therapy led to undetectable HBV DNA and normalization of liver enzymes. Even so, at the age of 55, after having been on anti-HBV therapy with undetectable HBV DNA for 5 years, the oldest son presented with chronic fatigue. Abdominal imaging revealed a 2.8 cm HCC. Despite transarterial chemoembolization (TACE) to the tumor and continuation of antiviral therapy, he progressed to requiring liver transplantation and was listed for one. Unfortunately, while awaiting liver transplantation he had HCC recurrence a year later and died within 12 months at the age of 56 (Figure 1).

The second son remained on antiviral therapy and HCC surveillance with abdominal ultrasound and alpha-fetoprotein every 6 months. After 8 years of successful antiviral therapy, at age 62, he was diagnosed with HCC. He underwent TACE but developed recurrence 3 months later.

The youngest child, the daughter, now 58 years old, has remained well on antiviral therapy with HCC surveillance and without HCC detection to date.

Currently, the mother and daughter continue to be on antiviral therapy and remain clinically well. The second son is listed for orthotopic liver transplantation (Figure 1).

### 2.2. Family 2

Early in 1990, during a routine examination, a 53-year-old woman was found to be positive for HBsAg while her husband was positive for hepatitis B surface antibody (anti-HBs). Soon thereafter, her three sons (aged 26, 28, 30) were tested and found to be HBsAg positive. The mother and her three sons had HBV genotype C, suggesting perinatal infection. None of them had co-infections with HIV or HCV. The family was then lost to follow up. Sixteen years later, at age 44, the middle-born son presented with chronic fatigue and was found to have a 6 cm HCC on abdominal imaging. Despite surgical resection and initiation of antiviral therapy with TDF, he died within 6 months of HCC diagnosis (Figure 2).

Soon, his two brothers, one older and the other younger, then established care at our institution. They were found to have elevated alanine aminotransferase (ALT) as well as HBV DNA without liver cirrhosis or HCC and, were subsequently started on antiviral therapy with TDF as well. Antiviral therapy led to undetectable HBV DNA levels in the blood and normalization of liver enzymes. They have done well the past 15 years since their first visit to our institution (Figure 2). Of note, the mother and her children also denied significant alcohol consumption.

Their mother, now 30 years after the first diagnosis of HBV infection and at age 83, has remained an asymptomatic HBV carrier with serologies as follows: HBsAg positive, anti-HBs negative, hepatitis B e antigen (HBeAg) negative, and low HBV DNA levels. She also has normal ALT levels and normal liver ultrasounds. She has never received HBV therapy due to the lack of indication for treatment.

The offspring of the children in family 1 and family 2 are fully vaccinated for HBV and have screened negative for HBV.

## 3. Discussion

Presented here is the longitudinal observations, over multiple decades, of two family clusters affected by vertically transmitted HBV that highlight the gender disparity associated with HBV-related HCC. Not only were more males affected by HCC, but the affected males also had more severe manifestations, including early age at diagnosis, death, and, requiring liver transplantation. In family 1, despite having 3 sequential HCCs, the mother has remained well for over 20 years since her HCC treatment. Her eldest son, however, succumbed to the HCC at the young age of 56 and the surviving son with recurrent HCC is listed for liver transplantation. Similarly, in family 2, of the three sons, only the middle-born son died of HCC. The two other sons are on antiviral therapy while their mother has remained well as an asymptomatic HBV carrier for over 30 years with no indication for antiviral therapy. Not many reports are available on longitudinal observations of family clusters with CHB, but of the ones reviewed, similar observations can be seen [2].

Vertical transmission is a common form of acquiring HBV [2]. In the setting of vertically transmitted HBV, the mother and child possess genetically identical viral genotypes while having similar host genomes [2]. Despite this, as shown here, it has been documented in such families that clinical outcomes from CHB can vary widely, ranging from lifelong asymptomatic infection to terminal HCC [1]. Of note, these observations occurred with the achievement of the treatment goal with targeted anti-HBV therapy [1].

The public health burden of hepatitis B underlines the importance of highlighting and understanding the constellation of risk factors that predispose, as well as affect prognosis of, chronically infected individuals to HCC [1]. It is well established that HCC predominantly affects males, with incidences two to four times more common than that in females, which remains true for HBV-related HCC [6,17,18,22]. Below we explore the published literature on the role of gender disparity not just as a risk for HBV-related HCC but as a prognostic indicator.

In a prospective cohort study from Hawaii involving 1206 patients who developed HCC, though not limited to HBV-related HCC, males developed HCC at a younger age, had higher MELD scores, and had larger tumor size(s) [17]. Similar observations were made in a prospective Italian study involving 1834 patients that aimed to ascertain whether female HCC patients have a better prognosis [21]. Notably, their study found that female HCC patients were at an older age at diagnosis, more likely to be diagnosed earlier and via surveillance as opposed to symptoms and, typically had smaller and more well differentiated tumors as well as a significantly longer survival (median 29 (95% confidence interval (CI): 24–33) vs. 24 (22–25) months, *p* = 0.0001). Interestingly, in this study, female HCC patients were more likely to present with increased levels of alpha-fetoprotein, though they had a better survival. The study by Yang et al. also reported a statistically significant survival advantage in women compared to men with HCC, even in the absence of major differences in disease burden or broad treatment categories [18].

The physiologic mechanisms underlying why males are more susceptible to developing HCC after an HBV infection, and have less favorable prognosis is an important yet widely unexplored topic [21], especially in the United States. This gender disparity is thought to be complex, multifactorial, and likely arise from gender differences in behavioral risk factors, hormonal factors, metabolic factors, and tumor biology [17]. Epigenetic and genetic alterations have also been implicated in the gender disparity in HBV-related HCC, but the exact mechanisms remain largely unknown and require further investigations [6]. These factors have been extensively reviewed elsewhere and summarized below [1,2,5,6,17,18,21,23].

### 3.1. Role of Behavioral Risk Factors in the Gender Disparity in HBV-Related HCC

Observational studies have found that females are more likely to be compliant with HCC surveillance guidelines as recommended by the American Association for the Study of Liver Diseases (AASLD) and European Association for the Study of the Liver (EASL) [24]. In turn, HCC surveillance is associated with significant improvements in early tumor detection, receipt of curative therapy, and overall survival in patients with cirrhosis [24].

A growing number of studies also so support the existence of sex differences in several aspects of consumption of alcohol, an established hepatotoxin and hepatocarcinogen, as well as alcohol use disorder (AUD) [25]. AUD is more prevalent in men [25,26,27]. Additionally, men have been reported to be more likely to drink alcohol excessively compared to women [25,27,28]. Furthermore, in the United States, men are more likely to have an alcohol-related hospitalization and death secondary to excessive drinking [27].

### 3.2. Role of Chronic Inflammation in the Gender Disparity in HBV-Related HCC

The pathophysiology of the gender disparities involved in HBV hepatocarcinogenesis, however, in most of the implicated risk factors stem from their gender differences in triggering chronic inflammation. Chronic inflammation is a known major contributor to tumorigenesis [1,2,6]. The pathogenesis of HBV-related HCC is ultimately attributed to a persistent inflammatory state created by HBV infection, which indirectly leads to the accumulation of alterations in the host genome that alas confer cell growth advantage [1,29].

Hepatitis-related inflammatory cytokines, such as interleukin-6 (IL-6) and interleukin-1β (IL-1β), have been identified as core promoters of susceptibility to an HBV infection, persistence of an HBV infection, and the initiation, promotion, and progression of the development of HBV-related HCC [2,6,29]. Studies have suggested that the effects of chronic inflammation and inflammatory cytokines on the development and progression of HBV-related HCC may vary in patients partly due to their gender difference, with less favorable outcomes seen in males [6]. This has been supported by studies involving administration of chemical carcinogens, such as diethylnitrosamine (DEN) and carbon tetrachloride (CCl4), to mice. Gender disparity is seen in mice given DEN [29] and CCl4 [6], where more apoptosis, necrosis, and compensatory proliferation of hepatic cells was induced in male mice than in female mice.

### 3.3. Role of Stress in the Gender Disparity in HBV-Related HCC

Chronic stress, as a risk factor for HBV-related HCC specifically, has been discussed extensively and thought to be potentially related to the immunologic and inflammatory reactions incited by the chronic stress [2,30,31]. Furthermore, a study has demonstrated strong inflammatory profiles due to psychological stress by demonstrating that the patients’ severity of depression independently correlated with higher levels of pro-inflammatory markers (e.g., IL-6, NF-κB) after undergoing induction of acute stress via the Trier Social Stress Test [32]. On the contrary, an interesting report attributed the gradual regression of a Liver Imaging Reporting and Data System (LI-RADS) 4 (probably hepatocellular carcinoma) lesion to a LI-RADS 1 (favor benignity) lesion in a patient with cirrhosis from HBV not just to sustained control of viral replication but to diminished stress levels as well [30]. Gender differences have been identified in the inflammatory response to stress, with an exaggerated inflammatory response to stress being noted in males compared to females [32].

### 3.4. Role of Metabolic Factors in the Gender Disparity in HBV-Related HCC

Gender disparities have also been noted in metabolic risk factors for HCC. Diabetes and obesity have been linked with excretion of inflammatory cytokines and therefore can cause hepatic inflammation and oxidative stress resulting in hepatocyte injury and, subsequently HCC [5]. Previous reports found that among men with diabetes, the risk of chronic nonalcoholic liver disease and HCC is doubled compared to their female counterparts [5].

### 3.5. Role of Sex Hormones in the Gender Disparity in HBV-Related HCC

Lastly, the role of sex hormones in HBV-related HCC has attracted great attention. The gender bias in the risk of and progression of HBV-related HCC has generated significant interest in the scientific community, especially the role of sex hormones in hepatocarcinogenesis [6,18,21]. Hormonal status is known to be important in modulating the risk of liver cancer [21,22]. An increasing number of studies have suggested that HBV-associated HCC may be a hormone-responsive malignant tumor [6] and various studies have explored the mechanisms involved. It has been reported that high levels of serum testosterone in males with HBV infection are associated with their development of HCC [20,33]. Androgens have demonstrated a synergistic oncogenic effect with HBV in men but not women [18]. The possible mechanisms through which androgens exert their effects is thought to be via activation of the androgen receptor (AR) [6] and its subsequent signaling pathways. Another hypothesized mechanism is via the Wnt signaling cascade, as evidenced by gene expression analyses and mouse models; however, further investigations are warranted to elucidate the exact pathways, especially as Wnt inhibitors enter into clinical practice [34]. Further supporting this is the presence of both higher androgen levels and more active androgen receptor gene alleles among male HBV carriers with an increased risk of HCC [6].

Unlike androgens, previous reports indicate that estrogen may play an important role in protecting against the development and progression of HBV infections, including the development of HBV-related HCC, by decreasing HBV RNA transcription and inflammatory cytokine levels [6,17,18,29,35]. It has been shown in mice studies that estrogens, at concentrations present in females but not in males, inhibit chemically induced hepatocarcinogenesis by suppressing IL-6 production [29]. A similar mechanism could account for the gender bias in liver cancer in humans [29]. Whether or not a disparity exists between the sexes in serum IL-6 levels or in the levels of expression of IL-6 mRNA in the hepatocytes of HBV-infected liver is not known, but it might be worth investigating [21].

Although a detailed review of the biologic impact of estrogen on HCC is beyond the scope of this article, at the mechanistic level, estrogen is thought to exert its protective effects through the estrogen receptor subtype alpha [18]. Estrogen exerts its effects through three estrogen receptors (ERs): ER-alpha (ERa), ER-beta (ERb), and G protein-coupled ER. Genome-wide expression and microRNA analyses in patients with HCC have shown ERa to be a tumor suppressor protein whose expression is inversely correlated with the presence of HBV infection, tumor size, and disease stage [18]. Reports of preclinical models reveal that loss of ERa accelerates the development of DEN-induced HCC by promoting hepatocyte necrosis over apoptosis [18]. Nonetheless, other studies have demonstrated that variant ER expression predominates in male patients with HCC and predicts worse survival, especially in those infected with HBV [6,18]. Expression of these variant receptors increases genomic oxidative damage, c-myc mRNA expression, and genomic instability, predisposing to carcinogenesis, and heralds ominous prognosis [21]. Another hypothesis is that estrogens repress HCC growth by inhibiting tumor-associated macrophages and preventing ERb from interacting with ATP5J, a part of ATPase, thus inhibiting the JAK1-STAT6 pathway [18].

Studies on therapies targeting the hormonal sensitivity of HCC have thus far shown mixed results. Hormonal replacement therapy has been related to a lower risk of developing HCC [18]. However, endocrine therapies using tamoxifen, a type of drug that exerts a relatively more estrogenic than anti-estrogenic effect in liver tissues, exhibited only modest beneficial effects in a relatively small population of patients with advanced HCC and sometimes resulted in a worse survival rate and increased negative impacts. Anti-androgenic drugs, such as flutamide, failed to provide a remarkable survival advantage when applied in hormone-blocking treatments for HCC and the use of hormonal drugs in patients with advanced HCC is not currently recommended [6].

### 3.6. Future Perspective

Further investigations to validate the aforementioned mechanisms, especially the expression of sex hormone receptors and understanding the signaling pathway they affect, may offer critical information regarding novel opportunities in gender-related personalized therapies for treatment and management of CHB and HBV-related HCC, for example, by emulating a previous study of HCV-related HCC that aimed to evaluate the gender-based (males vs females) ER subtype protein expression in the liver of normals (basal expression) and further evaluate the changes in the expression of these subtypes in HCV- related cirrhosis or HCV-related HCC patients [36]. The impact of the gender-associated ER subtype ratio distribution in the liver can also be studied relative to oncogenic markers and that may help us to monitor host therapeutic responses, viral clearance, or improved prognosis during the carcinogenesis process [36].

## 4. Conclusions

The importance of sex differences in outcomes of HBV-related HCC was highlighted by the observations in this case series, with females having a better prognosis. Increasing evidence shows that gender disparity is an important factor that strongly influences the risk of development and outcomes from HBV-related HCC, including mortality [6,17,21]. The observed disparity may depend on a series of different pathogenic determinants, including the levels of sex hormones, inflammatory cytokines, and epigenetic and genetic alternations [6]. The clinical practice/patient care applications of this, however, remain to be elucidated. In the meantime, it is important that clinicians and researchers alike consider the sex of their patients when caring for or designing and administering treatments chronic hepatitis B and HBV-related HCC because the outcomes will likely differ [22].

## Figures and Tables

**Figure 1 vaccines-09-00838-f001:**
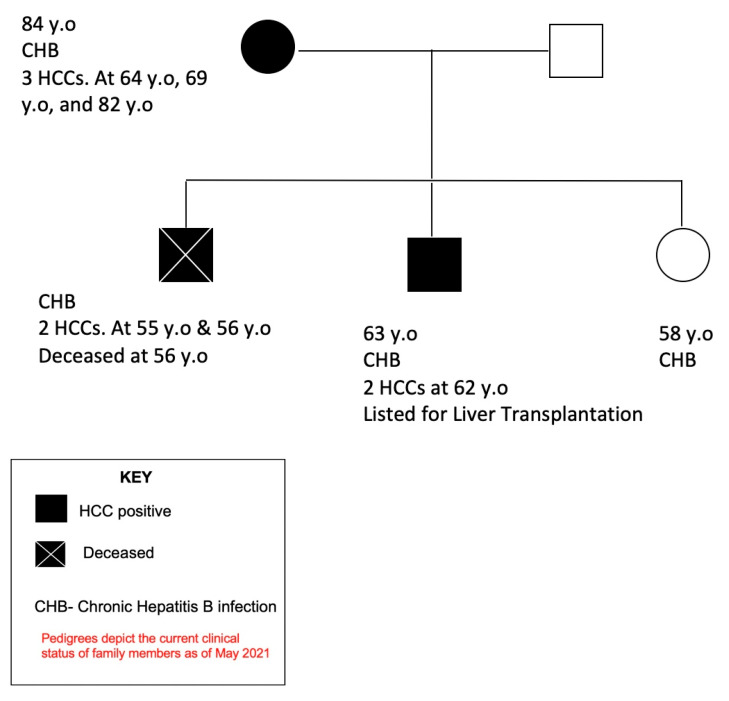
Pedigree of one family cluster (family 1) depicting the current clinical status of family members as of May 2021. Abbreviations: chronic hepatitis B (CHB), hepatocellular carcinoma (HCC), years old (y.o.).

**Figure 2 vaccines-09-00838-f002:**
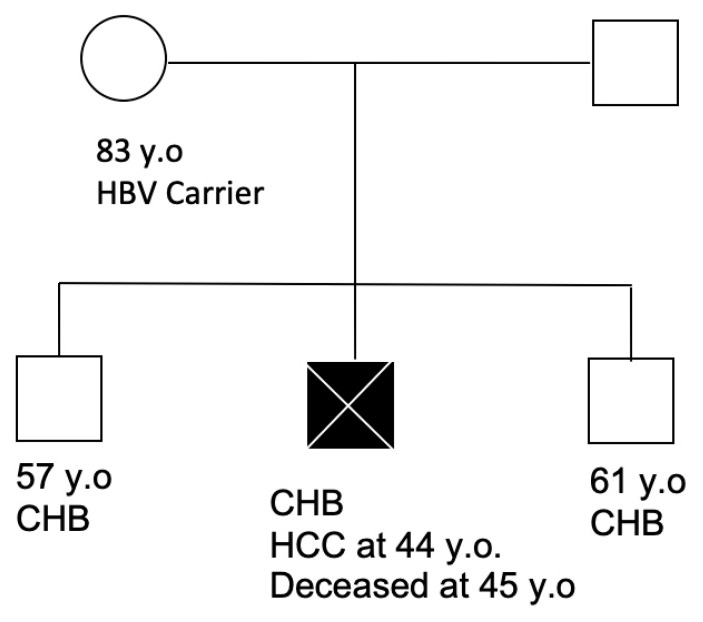
Pedigree of one family cluster (family 2) depicting the current clinical status of family members as of May 2021. Abbreviations: HBV (hepatitis B virus), chronic hepatitis B (CHB), hepatocellular carcinoma (HCC), years old (y.o.).

## Data Availability

The authors confirm that the data supporting the findings of this study are available within the article.

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
