# Peer review of "Gender Disparity in Host Responses to Hepatitis B-Related Hepatocellular Carcinoma: A Case Series"

_vaccines, 2021, doi:10.3390/vaccines9080838_

Round 1

Reviewer 1 Report

Dear Editor,

the authors described a case series with literature review of gender disparity in host responses to HBV-related HCC. According to my point of view the overall presentation of paper is good. The literature review about gender medicine is very important, the manuscript need to be publish. I have just some suggestions:

  • Introduction section: the authors missed to report one sentence about the importance of HBV vaccination. I suggest to insert one/two references about that (such as Stroffolini T eta al. 2012, doi:10.1016/j.ejim.2012.08.009.) or similar.
  • The authors should indicate the time period of downloaded papers to perform the literature review (from ... to ... ).
  • I suggest to improve discussion about published manuscript using the following paper from Ruggieri A et. al 2018 (doi:10.3389/fimmu.2018.02302).

Kind regards.

Author Response

Dear reviewer, thank you for your review and thoughtful comments. We have included a sentence about the importance of HBV vaccination (in the introduction section) and edited the discussion section as suggested as well.

The references added are:

Chang, Mei-Hwei. "Cancer prevention by vaccination against hepatitis B." Cancer Prevention II (2009): 85-94.

and the one suggested by you: Stroffolini T eta al. 2012, doi:10.1016/j.ejim.2012.08.009

The text:

Since the advent of the effective HBV vaccine and the widespread implementation of HBV vaccination programs, the incidence of HBV infection as decreased worldwide3, 4. Nonetheless, the burden of HBV infection persists; it roughly affects 2 billion people worldwide, among which approximately 300 million people are chronically infected1, 2. Chronic hepatitis B (CHB) is a common cause of both liver cirrhosis and hepatocellular carcinoma (HCC)1, 2, 5. Given its known carcinogenic properties, HBV accounts for 25% of HCC cases in developed countries and nearly 60% of cases in developing countries1, 2, 6. Clinical factors that determine outcomes of chronic HBV infection, and HCC, thus carry substantial public health importance2. Illumination of this constellation of risk factors, and understanding the HBV-related hepatocarcinogenesis remains an essential goal of HBV research1.

There has been significant progress in understanding viral chemistry-related hepatocarcinogenesis and as a result, advances have been made in targeted anti-HBV therapy which has ultimately decreased the morbidity and mortality associated with chronic HBV infection5. The incidence of HBV-related HCC has also decreased since the advent of HBV vaccine and antiviral therapy4, 7-10. Nonetheless, despite successful treatment of HBV infection with antiviral therapy, persistent risk for HCC has been reported11-16. A phenomenon that is also well documented at our high-volume hepatitis B center. This suggests that factors beyond genetic make-up of the virus, and therefore response to target therapy with antivirals, influence the outcome of CHB infection. On the contrary, host factors may play a larger role in the development and prognosis of HBV-related HCC. This idea has spiked wide interest in the scientific community2, 17.

Your sincerely, 

Rukaiya, Divya, and Dr. Hann

Reviewer 2 Report

This publication is undoubtedly interesting. Unfortunately I have one comment. I am asking for the contribiuton  of other factors more often see in man ( e.g toxic, metabolic ) which my be involved in HCC development.  

Author Response

Dear reviewer, thank you for your kind words and thoughtful comments. We included a section on the potential role of gender-disparity in metabolic risk factors that may also contribute to the gender disparity seen with HBV-related HCC. There is unfortunately a significant paucity of relevant data to include in regards to gender differences in possible hepatotoxins between males and females so it is only briefly touched upon here. We hope that this case series paper and the references included will potentially lend to future studies related to this important topic. 

From the text:

The physiologic mechanisms underlying why males are more susceptible to developing HCC after an HBV infection, and have less favorable prognosis is an important, yet widely unexplored topic21, especially in the United States. This gender disparity is thought to be complex, multifactorial and likely arise from gender differences in behavioral risk factors, hormonal factors, metabolic  factors and tumor biology17. Epigenetic and genetic alterations have also been implicated in the gender disparity in HBV-related HCC, but the exact mechanisms remain largely unknown and require further investigations6. These factors are extensively reviewed elsewhere and summarized below1, 2, 5, 6, 17, 18, 21, 23.

Role of metabolic factors in the gender disparity in HBV-related HCC

Gender disparities have also been noted in metabolic risk factors for HCC. Diabetes and obesity have been linked with excretion of inflammatory cytokines and therefore, can cause hepatic inflammation and oxidative stress resulting in hepatocyte’s injury, subsequently HCC5. Previous reports found that among men with diabetes, the risk of chronic nonalcoholic liver disease and HCC is doubled compared to their female counterparts5.

Best regards, 

Rukaiya, Divya, and Dr. Hann

Reviewer 3 Report

Authors introduced two family groups of HBV chronic infection and exhibited HCC incidence in part of family members and some were poor prognosis. They also discussed about the outcome of HBV infected patients related with sexual disparity, for example, the role of chronic inflammation, stress, or sexual hormones. However, there is insufficient matter about the relationship between case reports of two-family groups and author’s discussion.

Major comment.

Information of HBV infected family cases is insufficient. It should include essential information about the name of drugs for HBV antiviral therapies, degree of alcohol intakes, fibrosis status for CH-B patients, though listed LT indicated patients might be advanced liver fibrosis. If the antiviral therapy is nucleos(t)ide analogues, it is better to be listed the name. Also, should check the status of children of these patients so that it is described three generations, even if they are father, and even if they might be taken HBV vaccine.

Author Response

Dear reviewer, thank you for your thoughtful comments. We had used the style/level of details in previously published similar case series such as: Block, Peter, et al. "Vagaries of the host response in the development of hepatitis B-related hepatocellular carcinoma: a case series." Current Cancer Therapy Reviews 16.3 (2020): 253-258. We appreciate your feedback and in the revised copy, we have provided additional information regarding the families reported. 

From the text:

Family 1

In 1987, during a community HBV screening, an Asian mother and her three adult children (two older brothers and a younger sister) were found to be positive for hepatitis B surface antigen (HBsAg) with HBV genotype C, suggesting perinatal infection. None of them had human immunodeficiency virus (HIV) or hepatitis C virus (HCV) co-infections. Upon longitudinal observation, all four patients went on to develop chronic hepatitis B. Interestingly, their course of CHB-related illness over the following decades was observed to be different.

After 14 years without complications or antiviral therapy, in the year 2000 and at 64 years old (y.o), the mother developed fatigue and presented to our institution for a new patient visit. Further evaluation revealed an alpha-fetoprotein level of 6.7 (normal < 6.1 ng/mL), alanine aminotransferase (ALT) of 29 (normal < 45 IU/L), and an abdominal magnetic resonance imaging (MRI) with a 2 x 1.5 centimeter (cm) lesion consistent with HCC in the absence of evidence of cirrhosis or portal hypertension. She was started on antiviral therapy with Lamivudine, underwent microwave tumor ablation, and remained well with undetectable HBV DNA levels.  However, she went on to have two HCC recurrences (at the age of 69 and at the age of 82), both managed with loco-regional tumor ablation and continuation of antiviral therapy. Lamivudine was later switched to Tenofovir disoproxil fumarate (TDF). Thereafter, her 3 children established care at our institution. They were found to be in low viremic stages with minimal signs of hepatic inflammation, and antiviral therapy with TDF was started in 2008 in light of their mother’s multiple HCCs. Antiviral therapy led to undetectable HBV DNA and normalization of liver enzymes. Even so, at the age of 55, after having been on anti-HBV therapy with undetectable HBV DNA for 5 years, the oldest son presented with chronic fatigue. Abdominal imaging revealed a 2.8 cm HCC. Despite transarterial chemoembolization (TACE) to the tumor and continuation of antiviral therapy, he progressed to requiring liver transplantation and was listed for one. Unfortunately, while awaiting liver transplantation he had HCC recurrence a year later and died within 12 months at the age of 56 (Figure 1).

The second son remained on antiviral therapy and HCC surveillance with abdominal ultrasound and alpha-fetoprotein every 6 months. After 8 years of successful antiviral therapy, at age 62, he was diagnosed with HCC. He underwent TACE but developed recurrence 3 months later.

The youngest child, the daughter, now 58 years old has remained well on antiviral therapy with HCC surveillance and without HCC detection to date.

Currently, the mother and daughter continue to be on antiviral therapy and remain clinically well. The second son is listed for orthotopic liver transplantation (Figure 1).

Family 2

Early in 1990, during a routine examination, a 53-year-old woman was found to be positive for HBsAg while her husband was positive for hepatitis B surface antibody (anti-HBs). Soon thereafter, her three sons (aged 26, 28, 30) were tested and found to be HBsAg positive. The mother and her 3 sons had HBV genotype C, suggesting perinatal infection. None of them had co-infections with HIV or HCV. The family was then lost to follow up. Sixteen years later, at age 44, the middle-born son presented with chronic fatigue and was found to have a 6 cm HCC on abdominal imaging. Despite surgical resection and initiation of antiviral therapy with TDF, he died within 6 months of HCC diagnosis (Figure 2).

Soon, his two brothers, one older and the other younger, then established care at our institution. They were found to have elevated alanine aminotransferase (ALT) as well as HBV DNA without liver cirrhosis or HCC and, were subsequently started on antiviral therapy with TDF as well. Antiviral therapy led to undetectable HBV DNA levels in the blood and normalization of liver enzymes. They have done well the past 15 years since their first visit to our institution (Figure 2).

Their mother, now 30 years after the first diagnosis of HBV infection and at age 83, has remained an asymptomatic HBV carrier with serologies as follows: HBsAg positive, anti-HBs negative, hepatitis B e antigen (HBeAg) negative and low HBV DNA levels. Also, she remains with normal ALT levels and normal liver ultrasounds. She has never received HBV therapy due to the lack of indication for treatment.

Thank you.

Best regards, 

Rukaiya, Divya, and Dr. Hann

Round 2

Reviewer 3 Report

As commented at first review, it should be included HBV status in three generations.

Family 1; Are there children in “58 y.o CHB” female case? Describe their HBV status.

Author Response

Dear reviewer, 

Dr. Hann has double checked with the families and found that their offspring are all fully vaccinated for HBV and HBV negative. We have included that information under the write-up for the families as an additional sentence.

Thank you.

Best regards, 

Rukaiya, Divya, and Dr. Hann